# SARS-CoV-2 infection: Initial viral load (iVL) predicts severity of illness/outcome, and declining trend of iVL in hospitalized patients corresponds with slowing of the pandemic

Said El Zein[1], Omar Chehab[1◦], Amjad Kanj[2◦], Sandy Akrawe[1], Samer Alkassis[1], Tushar Mishra[1], Maya Shatta[1], Nivine El-Hor[1], Hossein Salimnia[3], Pranatharthi Chandrasekar[4]*

**1** Department of Internal Medicine, Detroit Medical Center, Wayne State University, Detroit, Michigan, United States of America, **2** Division of Pulmonary and Critical Care Medicine, Mayo Clinic, Rochester, Minnesota, United States of America, **3** Department of Pathology, Detroit Medical Center, Wayne State University, Detroit, Michigan, United States of America, **4** Division of Infectious Diseases, Department of Internal Medicine, Wayne State University School of Medicine, Detroit, Michigan, United States of America

◦ These authors contributed equally to this work.
* pchandrasekar@med.wayne.edu

**Data Availability Statement:** The data underlying this study are available publicly at https://doi.org/10.6084/m9.figshare.15085656.v3.

## Abstract

### Background

Hospitalization of patients infected with the severe acute respiratory syndrome virus 2 (SARS-CoV-2) have remained considerable worldwide. Patients often develop severe complications and have high mortality rates. The cycle threshold (Ct) value derived from naso-pharyngeal swab samples using real time polymerase chain reaction (RT-PCR) may be a useful prognostic marker in hospitalized patients with SARS-CoV-2 infection, however, its role in predicting the course of the pandemic has not been evaluated thus far.

### Methods

We conducted a retrospective cohort study which included all patients who had a nasopharyngeal sample positive for SARS-CoV-2 between April 4 –June 5, 2020. The Ct value was used to estimate the number of viral particles in a patient sample. The trend in initial viral load on admission on a population level was evaluated. Moreover, patient characteristics and outcomes stratified by viral load categories were compared and initial viral load was assessed as an independent predictor of intubation and in-hospital mortality.

### Results

A total of 461 hospitalized patients met the inclusion criteria. This study consisted predominantly of acutely infected patients with a median of 4 days since symptom onset to PCR. As the severity of the pandemic eased, there was an increase in the percentage of samples in the low initial viral load category, coinciding with a decrease in deaths. Compared to an initial low viral load, a high initial viral load was an independent predictor of in-hospital mortality

**Funding:** The author(s) received no specific funding for this work.

**Competing interests:** The authors have declared that no competing interests exist.

(OR 5.5, CI 3.1–9.7, p < 0.001) and intubation (OR 1.82 CI 1.07–3.11, p = 0.03), while an initial intermediate viral load was associated with increased risk of inpatient mortality (OR 1.9, CI 1.14–3.21, p = 0.015) but not with increased risk for intubation.

## Conclusion

The Ct value obtained from nasopharyngeal samples of hospitalized patients on admission may serve as a prognostic marker at an individual level and may help predict the course of the pandemic when evaluated at a population level.

## Introduction

As of March 11, 2021, more than 110 million cases of the severe acute respiratory syndrome virus 2 (SARS-CoV-2) have been reported worldwide. The majority of cases have been reported from the United States with more than 500,000 deaths so far in the U.S alone [1]. Several studies evaluated the association between SARS-CoV-2 viral load (VL) and patient outcomes. Zheng S et al. demonstrated that patients with severe disease had late shedding peaks compared to patients with mild disease. The authors also reported that the virus could be isolated up to 29 days after symptom onset with declining VLs over time [2]. More recently, several studies demonstrated that a high SARS-CoV-2 viral load was an independent predictor of mortality in large hospital cohorts [3–6] however, trends in the initial viral load over time at a population level have not been evaluated thus far. Also, similar data from a predominantly African American population have not been reported.

We describe a steady downtrend in the level of initial SARS-CoV-2 VL detected in nasopharyngeal samples of hospitalized patients in Detroit, Michigan as the pandemic evolved coinciding with a decrease in the percent of deaths. Moreover, a higher initial viral load on presentation in our patient population was an independent predictor of in-hospital mortality and risk for intubation, supporting the results of previously published reports.

## Methods

### Data collection

We conducted a retrospective study which included all patients who had a nasopharyngeal swab sample positive for SARS-CoV-2 infection by the Cepheid GenExpert instrument system–Rapid Real Time Polymerase Chain Reaction (RT-PCR) at the Detroit Medical Center during April 4, 2020-June 5, 2020. Data on 990 nasopharyngeal swabs were available for analysis, however, only 461 symptomatic patients met the inclusion criteria for our study (Fig 1). All sample containers, swabs, transport and storage, nucleic acid extraction and PCR protocols were similar for all included patients. We included results of the initial PCR sample obtained on presentation and omitted any subsequent results from the same patient. Medical records through July 5, 2020 were reviewed and all-cause in-hospital mortality was recorded. Patients who were discharged in stable condition were presumed alive at 30-day follow-up. Weeks were divided into intervals of seven days starting from April 4, 2020. In Michigan, the first peak in newly diagnosed COVID-19 cases was observed on April 7, 2020. By the end of week 2 of our study, the number of newly recorded cases had decreased by half and continued to decline steadily thereafter [7]. After May 9, 2020, we noticed a significant drop in the number

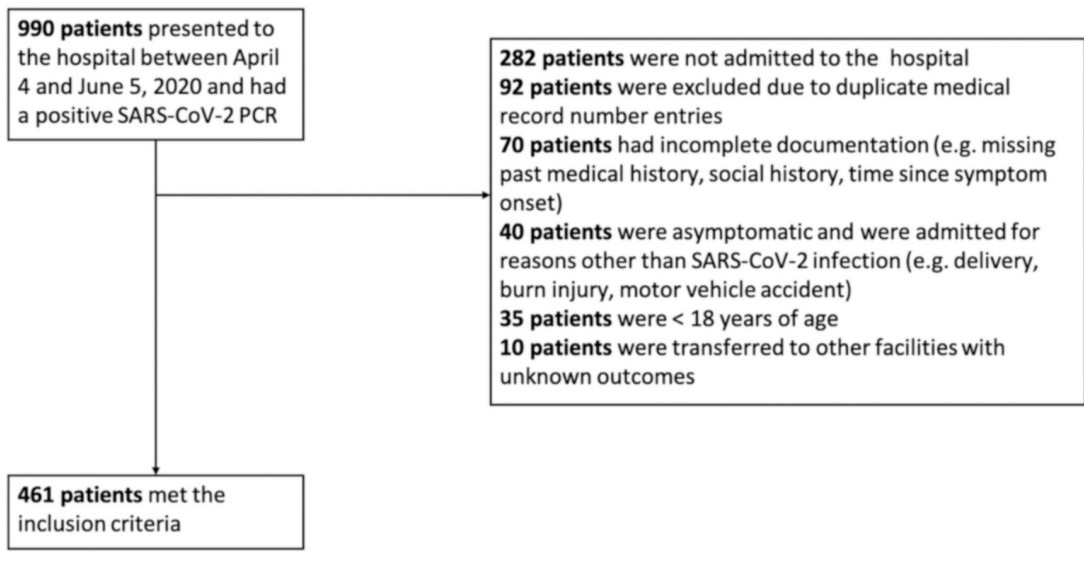

**Fig 1. Study flow diagram.**

of hospitalized SARS-CoV-2 cases, therefore, all cases between May 9, 2020 and June 5, 2020 were included in the 5+ week category.

## Viral load assessment

Samples were considered positive for SARS-CoV-2 when the N gene (nucleocapsid) was detected. Although the RT-PCR test is qualitative and results are reported as positive and negative, the Exprt Xpress SARS-CoV-2 assay reports the cycle threshold (Ct) when the test is positive [8, 9]. The Ct value can be used to estimate the number of viral particles in a patient sample. The assay used in this study does not have an internal control, however, it contains a Sample Processing Control (SPC) utilized by the GeneXpert Xpress System instrument to confirm adequate processing of the sample and to ensure that the RT-PCR reaction conditions are appropriate. In an ideal setting where the PCR efficiency is 100% and the R-squared value equals 1, a reduction of the Ct value by 3.3 is expected when the target concentration increases by one $\log_{10}$ [9]. Using a commercial quantified standard, our data showed a linear relationship between viral load and Ct values over four "1 to 10" dilutions of standard with an R-squared value of 0.99 (S1 Fig). At $2x10^2$, $2x10^3$ and $2x10^4$ viral particles, the Ct values were 43, 39.4 and 36.7 respectively. We designated high, intermediate, and low VL samples to have a Ct value of $\leq 25$, 26–36, and $\geq 37$ consecutively. Assuming a linear relationship between the Ct value and target concentration, samples with a Ct value of 26 should have a VL of $2x10^7$ while samples with a Ct value of 36 should have a VL of $2x10^4$ approximately. The lower limit of detection of our assay is around 250 genome copies/mL (95% confidence).

## Statistical analysis

Baseline characteristics and patient outcomes were stratified by viral load category. Medians and interquartile range (IQR) were used to describe continuous variables, while categorical variables were represented as proportions. The Kruskal-Wallis ANOVA test was used to compare medians, and the Fisher's exact test was used to compare categorical variables. A 2-sided $p$ value $\leq 0.05$ was used to designate statistical significance. A time-based analysis using Cox proportional hazards was used to compare the cumulative risk of in-patient mortality and

intubation among patients in the different viral load categories. Baseline factors that were associated with an increased risk for in-hospital mortality and intubation were identified on univariate analysis. All variables that were associated with each outcome having a *P* value of ≤ 0.1 were included in a multivariate logistic regression model (S1 Table) and adjusted odds ratios (OR) were calculated for these variables. Finally, we performed a multivariate logistic regression model using age and Ct values as continuous variables. Statistical analysis was completed using IBM SPSS Statistics version 26 (Armonk, NY: IBM Corp).

### Ethics statement

The Institutional Review Board at Wayne State University School of Medicine reviewed the study protocol and ethical approval for the conduction of this study was granted (IRB No. RR19393). The study was conducted under a waiver of informed consent. Data were analyzed anonymously.

## Results

### Declining trend in initial viral load over time

A total of 461 nasopharyngeal swabs were analyzed from hospitalized patients. During the first week of the study (week of April 4, 2020), the initial VL in the PCR-positive respiratory samples was predominantly in the intermediate group (46.3%, n = 57). As the pandemic evolved, there was a steady decline in the percent of positive samples in the high and intermediate VL categories with a concomitant rise in the percent of positive samples in the low VL category. By week five, 67% of the samples were in low VL category. This trend coincided with a decrease in the percentage of deaths in hospitalized patients (Fig 2).

### Patient characteristics stratified by viral load

Table 1 represents the baseline characteristics and outcomes of hospitalized patients with SARS-CoV-2 stratified by initial viral load category. The median age of patients with initial high, intermediate, and low viral load was 66, 69 and 71 years respectively (p = 0.002). Patients

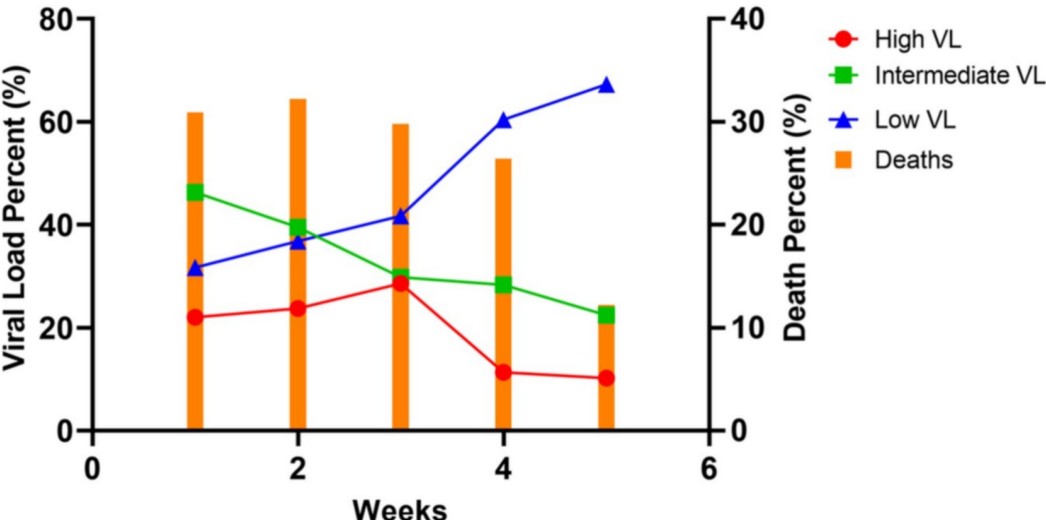

**Fig 2. Declining trend in the initial SARS-CoV-2 VL in hospitalized patients coinciding with a decrease in mortality rates.** Weeks were divided into intervals of seven days starting from April 4, 2020.

**Table 1. Baseline characteristics of hospitalized patients with SARS-CoV-2 infection stratified by initial viral load category.**

| Variable | Total | Low Viral Load | Intermediate Viral Load | High Viral Load | P Value |
|---|---|---|---|---|---|
| | n = 461 | (Ct $\geq$ 37) | (Ct 26–36) | (Ct $\leq$ 25) | |
| | | n = 195 | n = 168 | n = 98 | |
| I. Demographic Characteristics | | | | | |
| **Age (years), median (IQR)** | **68 (60–77)** | **66 (55–75)** | **69 (63–76)** | **71 (61–81)** | **0.002** |
| Age (by categories, years), No. (%) | | | | | |
| 18–39 | 22 (4.8) | 16 (8.2) | 1 (0.6) | 5 (5.1) | 0.001 |
| 40–65 | 166 (36) | 80 (41) | 53 (31.5) | 33 (33.7) | |
| > 65 | 273 (59.2) | 99 (50.8) | 114 (67.9) | 60 (61.2) | |
| Gender—No. (%) | | | | | |
| Men | 243 (52.7) | 110 (56.4) | 82 (48.8) | 51 (52.0) | 0.35 |
| Women | 218 (47.3) | 85 (43.6) | 86 (51.2) | 47 (48.0) | |
| Race—No. (%) | | | | | |
| African American | 362 (78.5) | 159 (81.5) | 126 (75) | 77 (78.6) | 0.41 |
| White | 38 (8.2) | 11 (5.6) | 17 (10.1) | 10 (10.2) | |
| Other | 61 (13.2) | 25 (12.8) | 25 (14.9) | 11 (11.2) | |
| **Nursing Home Resident** | **168 (36.4)** | **59 (30.3)** | **65 (38.7)** | **44 (44.9)** | **0.03** |
| II. Comorbidities—No. (%) | | | | | |
| Hypertension | 355 (77) | 146 (74.9) | 129 (76.8) | 80 (81.6) | 0.42 |
| Diabetes Mellitus | 203 (44.0) | 89 (45.6) | 70 (41.7) | 44 (44.9) | 0.73 |
| **Chronic Kidney Disease** | **128 (27.8)** | **65 (33.3)** | **46 (27.4)** | **17 (17.3)** | **0.013** |
| End Stage Renal Disease | 54 (11.7) | 27 (13.8) | 17 (10.1) | 27 (13.8) | 0.47 |
| Coronary Artery Disease | 99 (21.5) | 42 (21.5) | 32 (19) | 25 (25.5) | 0.46 |
| Heart Failure | 45 (9.8) | 17 (8.7) | 16 (9.5) | 12 (12.2) | 0.67 |
| Asthma | 31 (6.7) | 12 (6.2) | 11 (6.5) | 8 (8.2) | 0.80 |
| COPD | 83 (18) | 34 (17.4) | 29 (17.3) | 20 (20.4) | 0.87 |
| Smoking History | 168 (36.4) | 59 (30.3) | 65 (38.7) | 44 (44.9) | 0.57 |
| Autoimmune Disease | 24 (5.2) | 11 (5.6) | 9 (5.4) | 4 (4.1) | 0.84 |
| HIV | 7 (1.5) | 5 (2.6) | 1 (0.6) | 1 (1.0) | 0.28 |
| Hematologic malignancy | 8 (1.7) | 4 (2.1) | 2 (1.2) | 2 (2.0) | 0.82 |
| Solid Malignancy | 39 (8.5) | 17 (8.7) | 11 (6.5) | 11 (11.2) | 0.41 |
| Transplant Recipient | 15 (3.3) | 6 (3.1) | 6 (3.6) | 3 (3.1) | 0.95 |
| Immunosuppressive therapy | 19 (4.1) | 10 (5.1) | 7 (4.2) | 2 (2.0) | 0.08 |
| Body Mass Index (kg/m$^2$) | | | | | |
| < 18 | 11 (2.4) | 4 (2.1) | 6 (3.6) | 1 (1.0) | 0.13 |
| 18–25 | 123 (26.7) | 53 (27.2) | 40 (23.8) | 30 (30.6) | |
| 25–30 | 135 (29.3) | 53 (27.2) | 45 (26.8) | 37 (37.8) | |
| >30 | 192 (41.6) | 85 (43.6) | 77 (45.8) | 30 (30.6) | |
| III. Symptoms on presentation—No. (%) | | | | | |
| Gastrointestinal | 115 (24.9) | 53 (27.2) | 42 (25) | 20 (20.4) | 0.45 |
| **Respiratory** | **314 (69)** | **117 (60.6)** | **126 (75.9)** | **71 (74)** | **0.004** |
| Neurological | 162 (35.1) | 75 (38.5) | 50 (29.8) | 37 (37.8) | 0.18 |
| Systemic | 267 (57.9) | 107 (54.9) | 102 (60.7) | 58 (59.2) | 0.51 |
| **Days since symptom onset-to-PCR, median (IQR)** | **4 (2–8)** | **5 (2–12)** | **3 (2–7)** | **3 (1–5)** | **<.001** |
| IV. Laboratory variables (within 3 days of hospital admission), Median (IQR) | | | | | |
| Total White Blood Cell Count (x 10$^3$ cells/mm$^3$) | 9.5 (6.3–11.6) | 8.7 (5.8–12.4) | 8.3 (5.5–12.3) | 7.9 (5.4–11.5) | 0.64 |
| **Absolute Lymphocyte Count (x 10$^3$ cells/mm$^3$)** | **0.9 (0.5–1.2)** | **0.95 (0.55–1.3)** | **0.7 (0.4–1.0)** | **0.75 (0.47–1.2)** | **<.001** |

*(Continued)*

**Table 1.** (Continued)

| Variable | Total | Low Viral Load | Intermediate Viral Load | High Viral Load | P Value |
|---|---|---|---|---|---|
| | n = 461 | (Ct $\geq$ 37) | (Ct 26–36) | (Ct $\leq$ 25) | |
| | | n = 195 | n = 168 | n = 98 | |
| Absolute Neutrophil Count (x $10^3$ cells/mm$^3$) | 7.4 (4.3–9.8) | 6.8 (2.2–9.9) | 7.1 (5.1–10.5) | 8.2 (6.0–10.6) | 0.87 |
| **Platelets (x $10^3$ cells/mm$^3$)** | **226 (167–294)** | **235 (177–300)** | **234 (174–294)** | **202 (155–268)** | **0.02** |
| Creatinine (mg/dL) | 1.9 (1.0–4.3) | 1.5 (0.96–4.4) | 2.2 (0.97–6.0) | 2.0 (1.1–4.6) | 0.46 |
| Alanine Aminotransferase (units/L) | 20 (12–33) | 23 (11–31) | 19 (12.2–36.5) | 33.5 (16.2–57.5) | 0.31 |
| Aspartate Aminotransferase (units/L) | 35 (22–59) | 35 (22.25–72.5) | 40.5 (19–65.25) | 44.5 (32.7–75.2) | 0.04 |
| Alkaline Phosphatase (units/L) | 71 (57–95) | 76 (67–89) | 69 (52–97) | 69 (53–95) | 0.75 |
| **C-Reactive Protein (mg/L)** | **120 (60–161)** | **100 (40–150)** | **122 (74–167)** | **148 (67–209)** | **0.001** |
| **Ferritin (ng/mL)** | **507 (209–1277)** | **408 (172–1189)** | **512 (247–1097)** | **697 (261–1710)** | **0.01** |
| D-dimer (mg/L) | 2.07 (1.0–6.96) | 2.2 (1.0–6.7) | 1.83 (0.9–7.22) | 2.22 (1.0–6.99) | 0.69 |
| **LDH (units/L)** | **364 (252–542)** | **311 (230–424)** | **372 (260–609)** | **417 (244–688)** | **<.001** |
| Troponin (ng/L) | 28 (11–80) | 28 (9–97) | 29 (26–32) | 28 (24–35) | 0.71 |
| V. Vital Signs on Admission, median (IQR) | | | | | |
| Heart rate (beats per minute) | 93 (82–107) | 89 (82–102) | 98 (84–116) | 104 (90–115) | 0.09 |
| **Mean Arterial Pressure (mmHg)** | **90 (79–102)** | **92 (80–104)** | **89 (77–101)** | **87 (75–101)** | **0.04** |
| Temperature (°C) | 37 (36.6–37.7) | 36.9 (36.6–37.5) | 37.1 (36.6–38) | 36.9 (36.6–38) | 0.12 |
| VI. Chest X-ray Findings | | | | | |
| No Abnormal Findings | 84 (18.2) | 44 (26.2) | 22 (13.8) | 18 (19.4) | 0.07 |
| Unilateral Infiltrates | 93 (20.2) | 36 (21.4) | 35 (21.9) | 22 (23.7) | |
| Bilateral Infiltrates | 244 (52.9) | 88 (52.4) | 103 (64.4) | 53 (57) | |
| VII. Medications | | | | | |
| **Corticosteroids** | **302 (65.5)** | **102 (52.3)** | **123 (73.2)** | **77 (78.6)** | **< 0.01** |
| VIII. Clinical Outcomes | | | | | |
| **Highest Oxygen Delivery during hospitalization** | | | | | |
| No supplemental Oxygen Required | 73 (15.8) | 43 (22.1) | 23 (13.7) | 7 (7.1) | 0.01 |
| Non-Invasive Oxygen Delivery ^ | 267 (57.9) | 107 (54.9) | 102 (60.7) | 58 (59.2) | 0.01 |
| Intubation and Mechanical Ventilation | 121 (26.2) | 45 (23.1) | 43 (25.6) | 33 (33.7) | 0.13 |
| Days until intubation, Mean (IQR) | 2 (0–5) | 3.6 (2–5) | 2.3 (1–3) | 3.9 (2–5) | |
| **Shock Requiring Pressor support during hospitalization (n, %)** | **95 (20.6)** | **29 (14.9)** | **35 (20.8)** | **31 (31.6)** | **0.004** |
| **In-hospital Mortality (n, %)** | **132 (28.6)** | **31 (15.9)** | **50 (29.8)** | **51 (52)** | **<.001** |
| **Days until death, Median (IQR)** | **9 (4–16)** | **14 (9–15)** | **9 (6–10)** | **8 (2–5)** | **0.01** |

Normal laboratory range: Alanine Aminotransferase: 7–52 units/L; Aspartate Aminotransferase: 13–39 units/L; Alkaline Phosphatase: 45–115 units/L; C-reactive protein: <5 mg/L; Ferritin: 23.9–336.2 ng/mL; LDH: 140–271 units/L; High sensitivity Troponin: < 17 ng/L; D-dimer: < 0.5 mg/L.

^ Non-invasive oxygen delivery includes nasal canula, non-rebreather mask, face mask, high flow oxygen delivery.

with a higher initial viral load were more likely to be nursing home residents (45%, n = 44, p = 0.004). There was no significant association between initial viral load on presentation and gender, race, coronary artery disease, hypertension, obstructive lung disease, smoking history, heart failure, diabetes mellitus or body mass index (BMI). In this study, patients with high and intermediate initial viral loads had a median of 3 days from symptom onset to PCR testing compared to 5 days for patients with a low initial viral load. Patients with a higher initial viral load were more likely to require oxygen therapy by nasal canula, non-rebreather mask, or high-flow nasal canula as well as intubation and mechanical ventilation during their

hospitalization. They were also more likely to develop shock requiring pressor support and had lower mean arterial pressures on admission.

Patients with a high initial viral load were more likely to have lower absolute lymphocyte counts, higher lactate dehydrogenase (LDH), C-reactive protein (CRP) and ferritin levels on presentation. There were however no differences in the levels of troponin or chest X-ray findings on presentation among patients with high, intermediate, and low viral load respectively.

## Patient outcomes

The last day of study follow-up was on July 5, 2020. By that time, 28.6% (n = 132) of the patients had died during their hospital admission while 71% (n = 329) were alive, out of which 58.4% (n = 192) were discharged home and 41.6% (n = 137) were discharged to nursing homes, long-term acute care centers or rehabilitation centers. Patients with high initial viral loads were at increased risk for intubation and death during their hospital admission. Among patients with a high initial viral load, in-hospital mortality was 52% (n = 51), compared to 30% (n = 50) and 16% (n = 31) for patients with intermediate and low initial viral load respectively (P < 0.001). The risk for intubation and mechanical ventilation during hospital admission was 34% (n = 33) in patients with an initial high viral load compared to 26% (n = 43) and 24% (n = 45) in patients with an intermediate and low initial viral load respectively (P = 0.01). In a time-based analysis, an initial high viral load was associated with a hazard ratio (HR) of 5.1 (CI 3.2–8.1 p < 0.001) for in-hospital mortality and a HR of 1.6 (CI 1.03–2.5, p = 0.035) for intubation compared to a low initial viral load, whereas an intermediate initial viral load was associated with an increased risk for in-hospital mortality (HR 2.2, CI 1.4–3.5 p < 0.001) but not with increased risk for intubation (Figs 3 and 4).

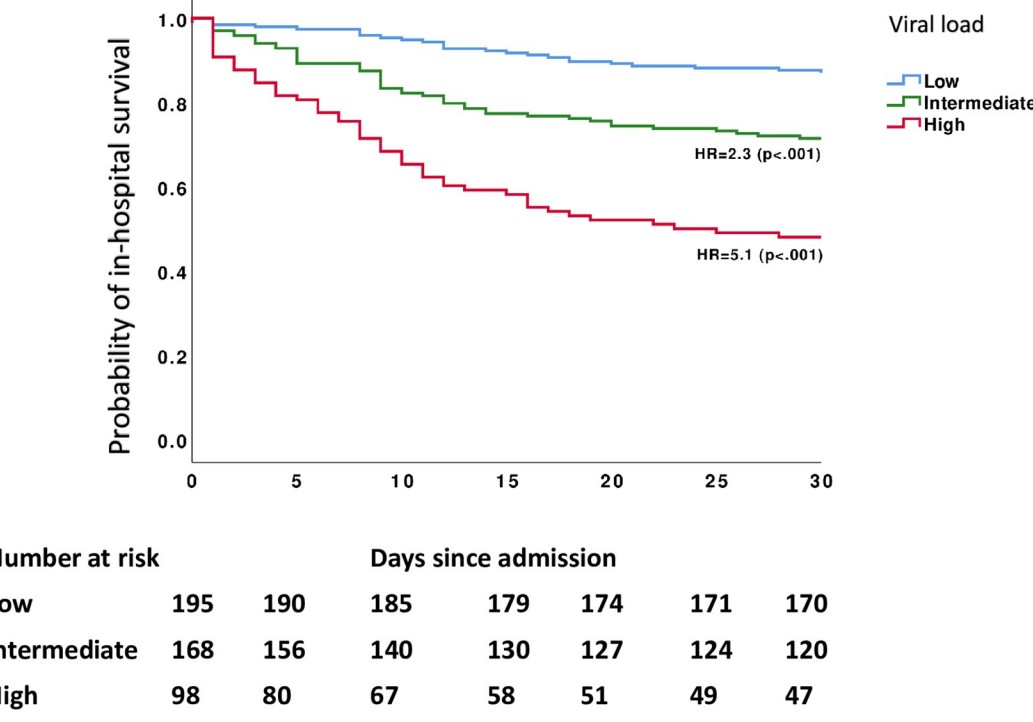

**Fig 3. Probability of in-hospital survival in patients with SARS-CoV-2 infection stratified by initial viral load category.**

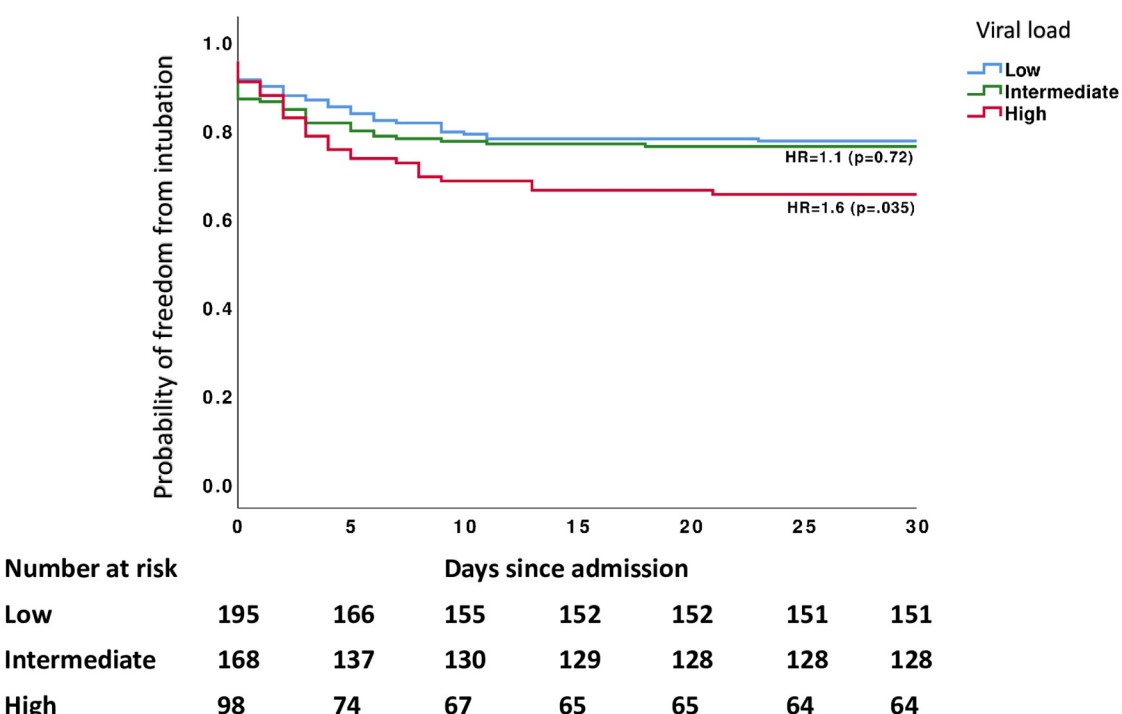

**Number at risk**

| | | | | | | | |
|---|---|---|---|---|---|---|---|
| Low | 195 | 166 | 155 | 152 | 152 | 151 | 151 |
| Intermediate | 168 | 137 | 130 | 129 | 128 | 128 | 128 |
| High | 98 | 74 | 67 | 65 | 65 | 64 | 64 |

**Fig 4. Probability of freedom from intubation in hospitalized patients with SARS-CoV-2 infection stratified by initial viral load category.**

Moreover, in a multivariate analysis adjusted to age and acute kidney injury, compared to a low initial viral load, both high and intermediate initial viral load were associated with increased risk of in-hospital mortality (OR 5.5, CI 3.1–9.7, p < 0.001 and OR 1.9, CI 1.14–3.21, p = 0.015) (Table 2). A high initial viral load was found to be associated with an increased risk

**Table 2. Multivariate analysis of factors associated with in-hospital mortality and intubation.**

| Variables | Mortality | |
|---|---|---|
| | Odds Ratio (95% CI) | p-value |
| **Viral Load By Nasopharyngeal Swab** | | |
| Low viral load (Ct $\geq$ 37) | Reference | Reference |
| Intermediate viral load (Ct 26–36) | 1.91 (1.14–3.21) | .015 |
| High viral load (Ct $\leq$ 25) | 5.50 (3.11–9.73) | <.001 |
| **Age (years)** | | |
| 18–39 | Reference | Reference |
| 40–65 | 2.24 (0.47–10.70) | 0.312 |
| >65 | 5.42 (1.16–25.2) | 0.031 |
| **CKD** * | 0.87 (0.52–1.45) | 0.598 |
| | **Intubation** | |
| **Viral Load By Nasopharyngeal Swab** | | |
| Low viral load (Ct $\geq$ 37) | Reference | Reference |
| Intermediate viral load (Ct 26–36) | 1.07 (0.66–1.725) | 0.779 |
| High viral load (Ct $\leq$ 25) | 1.82 (1.07–3.11) | 0.028 |

*__CKD:__ Chronic Kidney Disease.

**Table 3. Multiple logistic regression of independent predictors of in-hospital mortality using age and viral load as continuous variables.**

| Variables | Mortality | |
|---|---|---|
| | Odds Ratio (95% CI) | p-value |
| Age | 1.035 (1.018–1.053) | <0.001 |
| Ct Value | 0.921 (0.894–0.949) | <0.001 |

Variables entered in the model: Age (years), Ct value and chronic kidney disease (yes/no).

for intubation and mechanical ventilation OR 1.82 (CI 1.07–3.11, p = 0.03) whereas an initial intermediate viral load was not.

Using multiple logistic regression, both age and Ct value were found to be significantly associated with mortality, such as every 1-year increase in age results in 3.5% increase in odds of death and every 1 unit increase in Ct value decreases the odds of death by 8%. The negative association between Ct and mortality reflects the fact that lower Ct values indicate higher viral load (Table 3).

Finally, we explored the use of three drugs: corticosteroids, tocilizumab and remdesivir. At the time of the study, there was not enough evidence to support the use of any of these medications in the management of SARS-CoV-2 infection. A total of 302 patients (65.5%) received corticosteroids, while only 20 (4.3%) and 2 (0.4%) received tocilizumab and remdesivir respectively. Corticosteroids were prescribed to 52.3% (n = 102) of patients with low viral load, 73.2% (n = 123) of patients with intermediate initial viral load and 78.6% (n = 77) of those with high initial viral load (Table 1). Patients who received corticosteroids had higher mortality rates compared to patients who did not (35.8% vs 15.1%, p < 0.011). Since corticosteroids were generally reserved for more severe cases, the increased prevalence of their use in higher viral load categories reflects the relationship between high viral load and increased mortality highlighted in this study.

## Discussion

A gradual decline in the number of patients with a high initial viral load over time, coinciding with a declining death rate, corresponded well with the lessening severity of the pandemic. Similar observations were documented by Clementi N et al. from Italy. The authors noted that the mean Ct value of positive SARS-CoV-2 samples collected during the month of April, which represented the peak of the pandemic, was significantly lower than the mean Ct value of positive samples collected in May, a time when the pandemic in Italy became less severe [10].

Mina M et al. suggested that a changing population distribution of SARS-CoV-2 Ct values can be used to predict the course of the pandemic [11]. The authors speculated that when the pandemic is on the rise, the majority of tested patients will be acutely symptomatic, therefore, their SARS-CoV-2 PCR tests will demonstrate a low Ct value compared to high Ct values during pandemic decline as the majority of the patients will be in the convalescent phase [11]. Moreover, rapid implementation of social distancing measures and the widespread use of facemasks may have contributed to "variolation ", which may have helped slow down the spread of the virus [12], although, a recent study by Masia et al. demonstrated that patients with a lower viral load were less likely to seroconvert [13]. A change in the virus may also explain this observation, however, no data are available to support this conclusion. Major limitations for the use of the Ct value in clinical practice stem from the variability of sample processing protocols and the use of different PCR platforms. In our study, all sample containers, swabs, transport and

storage, nucleic acid extraction and PCR protocols were similar for all included patients and all Ct values were reported using the same instrument.

The correlation between VL, severity of illness and viral shedding for respiratory viruses remains a topic of debate. In patients with influenza infection, higher VLs were detected in hospitalized patients compared to ambulatory patients, however, this was generally not associated with worse outcomes [14, 15]. In contrast, there is evidence that among patients infected with the Middle East respiratory syndrome coronavirus (MERS-CoV) and severe acute respiratory syndrome virus (SARS), a higher VL was an independent risk factor for death [16, 17]. This study consisted predominantly of acutely infected patients with a median of 4 days since symptom onset to PCR. We demonstrated that the Ct value of nasopharyngeal swab samples obtained on initial presentation can act as a predictor of patient outcomes during hospitalization. Many studies correlated a higher SARS-CoV-2 viral load with increased severity of illness [3, 5, 6, 10, 18, 19]. A recent systematic review of 18 studies also supported that lower Ct values are associated with worse outcomes [20], and this study adds to the growing body of medical knowledge that a high initial nasopharyngeal viral load is an independent predictor of in-hospital mortality and intubation.

Higher plasma SARS-CoV-2 viral load may be associated with increased risk of death and higher levels of inflammatory markers such as lower lymphocyte count, higher C-reactive protein (CRP), and interleukin-6 levels [4]. We demonstrated that a higher initial nasopharyngeal viral load is associated with elevated ferritin and CRP levels as compared to low viral load on presentation. This raises the question of whether stratifying patients by levels of viremia and severity of inflammation can aid in determining which patients may benefit more from antiviral therapy versus anti-inflammatory therapy [4]. It remains unclear whether nasopharyngeal Ct values correlate closely with levels of plasma viremia as no quantitative SARS-CoV-2 PCRs received U.S Food and Drug Administration emergency use authorization, making their use limited to select centers with test availability. Proving a strong correlation between nasopharyngeal Ct values and levels of viremia may further support the use of the nasopharyngeal Ct as a prognostic marker for hospitalized patients with COVID-19 infection.

Despite comparable pandemic conditions, the 30-day in-hospital mortality in this study was 28.6%, which is significantly higher than the mortality rate reported by a similar cohort study by Magleby et al. from New York (19.2%) which included a total of 678 SARS-CoV-2 infected patients between March 30 –April 30, 2020 [3]. It is worth noting that both studies spanned a period when Michigan and New York city witnessed their first peak in newly diagnosed COVID-19 cases coinciding on April 7 and April 12 respectively [7]. Reasons for higher mortality seen in our cohort may be related to differences in underlying comorbidities, age, and decreased access to healthcare in the Detroit population. Notably, African Americans comprised 78.5% of our patient population, compared to 14% as reported by Magleby et al. [3] however, ethnicity itself was not an independent predictor of mortality in either study. Moreover, recent studies investigated the impact of race, socioeconomic and insurance status on COVID-19 outcomes in Detroit and found no association between these factors and COVID-19 mortality [21, 22]. The authors however reported that advanced age and comorbidities on admission were associated with more adverse outcomes [21, 22]. Remarkably, Tejada C et al. recently reported that all patients in a cohort of 25 African American renal transplant recipients who were hospitalized for COVID-19 had full recovery despite the presence of comorbidities and seemingly worse prognosis [23]. This was attributed to the use of steroids and calcineurin inhibitors which may have had an inhibitory effect on immune cell activation and concomitant cytokine dysregulation [23]. Rather than attributing higher death rates to intrinsic biologic susceptibilities, improving access to healthcare and managing comorbidities may eliminate ethnic differences in COVID-19 mortality [24, 25].

This study has some limitations. First, this is a retrospective cohort study, therefore, patients were not followed-up after discharge to document deaths that may have occurred outside the hospital. Moreover, given the retrospective nature of the study, some data may have been missed or misclassified during extraction from medical records, however, we performed constant queries during data abstraction to guarantee the accuracy of the data. Secondly, only the Cepheid Xpert Xpress SARS-CoV-2 PCR was used, therefore, our results may not be generalizable to all PCR platforms. Lastly, our data was obtained prior to the availability of vaccines and prior to the emergence of new variants.

In this study, we report an independent association between initial VL and risk for intubation and mortality in a cohort composed predominantly of SARS-CoV-2 infected African American patients. Our data suggest that the initial Ct value reported on hospital admission may serve as an important risk stratification tool by allowing physicians to identify patients who are at increased risk of developing adverse outcomes secondary to COVID-19 infection. Importantly, trends in the initial cycle threshold values over time may serve as a useful marker to assess the tempo or the direction of the pandemic in the region and may have an impact on public health measures, infection control and clinical management of infected patients.

## Supporting information

**S1 Table. Univariate analysis of factors associated with increased risk for intubation and mortality.**
(DOCX)

**S1 Fig. Standard curve.**
(TIF)

**S2 Fig.**
(PNG)

## Acknowledgments

Data collection team: Siri Sarvepalli, Vichar Trivedi, Adnan Halboni, Joseph Sebastian, Ishita Datta, Shiva Bongu, Zachary Cantor, Jiping Zhou, Jennifer Young, Jana Dbaibou, Bhavana Tetali, Tyler Mumm.

## Author Contributions

**Conceptualization:** Said El Zein, Pranatharthi Chandrasekar.

**Data curation:** Said El Zein, Omar Chehab, Amjad Kanj, Samer Alkassis, Tushar Mishra, Maya Shatta, Nivine El-Hor.

**Formal analysis:** Omar Chehab, Amjad Kanj.

**Investigation:** Omar Chehab, Hossein Salimnia, Pranatharthi Chandrasekar.

**Methodology:** Amjad Kanj, Hossein Salimnia, Pranatharthi Chandrasekar.

**Project administration:** Said El Zein, Hossein Salimnia, Pranatharthi Chandrasekar.

**Resources:** Hossein Salimnia, Pranatharthi Chandrasekar.

**Software:** Omar Chehab, Amjad Kanj.

**Supervision:** Hossein Salimnia, Pranatharthi Chandrasekar.

**Validation:** Amjad Kanj, Sandy Akrawe, Hossein Salimnia, Pranatharthi Chandrasekar.

**Visualization:** Hossein Salimnia, Pranatharthi Chandrasekar.

**Writing – original draft:** Said El Zein, Sandy Akrawe, Pranatharthi Chandrasekar.

**Writing – review & editing:** Said El Zein, Omar Chehab, Amjad Kanj, Sandy Akrawe, Samer Alkassis, Tushar Mishra, Maya Shatta, Nivine El-Hor, Pranatharthi Chandrasekar.

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
