## [Decision Letter · Decision Letter 0]

5 May 2021

PONE-D-21-08808

SARS-CoV-2 infection: Initial viral load (iVL) predicts severity of illness/outcome, and declining trend of iVL in hospitalized patients corresponds with slowing of the pandemic

PLOS ONE

Dear Dr. Chandrasekar,

Thank you for submitting your manuscript to PLOS ONE. After careful consideration, we feel that it has merit but does not fully meet PLOS ONE’s publication criteria as it currently stands. Therefore, we invite you to submit a revised version of the manuscript that addresses the points raised during the review process.

We look forward to receiving your revised manuscript.

Kind regards,

Tai-Heng Chen, M.D.

Academic Editor

PLOS ONE

Journal Requirements:

2) PLOS requires an ORCID iD for the corresponding author in Editorial Manager on papers submitted after December 6th, 2016. Please ensure that you have an ORCID iD and that it is validated in Editorial Manager. To do this, go to ‘Update my Information’ (in the upper left-hand corner of the main menu), and click on the Fetch/Validate link next to the ORCID field. This will take you to the ORCID site and allow you to create a new iD or authenticate a pre-existing iD in Editorial Manager. Please see the following video for instructions on linking an ORCID iD to your Editorial Manager account: https://www.youtube.com/watch?v=_xcclfuvtxQ

3)  Thank you for including your ethics statement:  "The Institutional Review Board reviewed the study protocol and ethical approval for the conduction of this study was granted (IRB No. RR19393). The study was conducted under a waiver of informed consent. Data were analyzed anonymously".   

4) We noted in your submission details that a portion of your manuscript may have been presented or published elsewhere. [poster presented at the ESCMID Conference]. Please clarify whether this [conference proceeding or publication] was peer-reviewed and formally published. If this work was previously peer-reviewed and published, in the cover letter please provide the reason that this work does not constitute dual publication and should be included in the current manuscript.

Reviewers' comments:

Reviewer's Responses to Questions

**Comments to the Author**

1. Is the manuscript technically sound, and do the data support the conclusions?

Reviewer #1: Yes

Reviewer #2: Partly

Reviewer #3: Yes

2. Has the statistical analysis been performed appropriately and rigorously? 

Reviewer #1: Yes

Reviewer #2: No

Reviewer #3: I Don't Know

3. Have the authors made all data underlying the findings in their manuscript fully available?

Reviewer #1: Yes

Reviewer #2: No

Reviewer #3: Yes

4. Is the manuscript presented in an intelligible fashion and written in standard English?

Reviewer #1: Yes

Reviewer #2: Yes

Reviewer #3: Yes

5. Review Comments to the Author

Reviewer #1: The present study was enrolled 461 SARS-CoV-2 infected patients who were admitted in hospital between April 4- June 5, 2020 to analyze the effect of initial viral load on the intubation and mortality of the patients, and drew the conclusion that low initial viral load was related to less chances of intubation and mortality, and coinciding with the descending trend of the pandemic. The data was analyzed with different but appropriate statistic ways in accord with their types, while the difference among the groups were given strong supports to the conclusion of the study. The article was written in a formal way but easy for understanding. All the tables and graphs are neat and well explained, only the image resolution is needed to get improved, especially Fig 3. Moreover, there is a missing part of sign “A” in Fig 3.

Reviewer #2: Said El Zein et al have studied the correlation of nasopharyngeal SARS-CoV-2 viral load and a series of factors ( signs, comorbidities, outcome...) of 461 hospitalized patients during April- June 2020. These patients were mostly of one ethnicity ( African-Americans) which makes the study more interesting. They found that The viral load ( stratified in 3 categories) is independently correlated with the outcome of the patients.

The subject is interesting, the number of cases studied is high enough, the findings are reasonable. However, I have the following comments and questions.

1- I am not familiar with the RT-PCR kit used in this study, but most of the kits that I have evaluated use internal control (IC) genes too. The Ct value of IC is a good indicator of the quality of the biological samples ( with respect to collecting enough biological material as well as lacking PCR inhibitors). It is more reasonable to consider the difference between the Ct of IC and SARS-CoV-2 gene than considering the latter as an indicator of viral load. Even if the authors prefer to use absolute viral load as their main variable, comparing the IC Cts may still be helpful to exclude any variation or change in sample collection technique during the this study, which may be another reason why the VLs declined over time.

2- using Ct as a correlate to the viral load needs more validation. Basically, as it is well mentioned by the authors, these kits have been developed as qualitative tests. Using just 2 standard points ( in this case 10000 and 100000 VPs) is not enough to use the results as an even semi-quantitative test. I would suggest using at least 4 standard concentrations of the virus to see if the test is linear and efficient in a broader range. It is note worthy that the observed difference between the 2 standard concentration is not 3.3 ( as it is written in the manuscript) but it is 3.6 which might have larger impacts in lower or higher viral loads.

3- It is not clear to me how the standard virus sample has been quantified. Does it come from a virus culture or a clinical sample? Is it quantified based on genetic material copy number or based on infectious particle counting by one of virus titration methods?

4- Following the previous comments on reliability of the quantification method, in my opinion choosing the cutoffs of 26 and 36 are not well justified. We have found several COVID cases with Ct values below 20. On the other hand the intercept of the standard curve of the majority of QPCR kits is slightly higher than 35 ( usually below 40), so the readouts above 35 are not usually reliable, or could be due to non-infectious viral debris.

5- Based on the previous comments and uncertainties regarding the QPCR results, I would recommend statistical re-analyzing the data with parametric test as well. In those analyses the absolute value of Ct ( or delta Ct, in case the Ct of the internal controls are available) could be used. Then the author may come to similar conclusion, or find out choosing different cutoffs for stratification is more suitable, or just report the results of the parametric analysis.

6-Based on results presented in table 1, it seems to me that Age and Viral load are rather highly correlated. In table 2 both have very close ORs on Mortality rate as well ( 5.50 vs 5.42). I wonder what is the correlation of these two variable with each other, and whether they are not covariates. Have the author tested other variables in different age groups ( or again, using age as a parametric variable against other variables)? In the Statistical Analysis section there is a reference to table S1, which I could not find it in the manuscript or its supplementary materials.

7- in table 1, the average level of Troponin is 40, while the average of each VL group is below 29. I wonder if it is correct.

Reviewer #3: The manuscript describes a cohort study on relation among initial viral load in nasopharyngeal covid-19 samples and severity of the disease as well as status of pandemic. The subject is well designed and performed although some points need authors’ attention:

1- An important factor influencing the outcome of the disease and the results of this study is the type of treatment and should be considered in all interpretations. Type of antiviral or anti-inflammatory drugs, interval time between PCR and starting the treatment should be considered in analysis.

2- An explanation of the division of weeks according to pandemic conditions should be given at the first appearance in the text.

3- In comparing with the report of Magleby et al. in addition to the reasons given for the discrepancy with the results of the present study, the authors should also examine and discuss the pandemic conditions in two different studied time periods. Similarly, due to the fact that pandemic peaks are not the same in different countries, this point should be considered in comparison with the data of other countries.

4 -The sentence in line 97 and 98 needs reference.

6. PLOS authors have the option to publish the peer review history of their article (what does this mean?). If published, this will include your full peer review and any attached files.

Reviewer #1: No

Reviewer #2: No

Reviewer #3: No

---

## [Author Response · Author response to Decision Letter 0]

18 May 2021

Dear editor

Tai-Heng Chen, M.D.

PLOS One 

Thank you for considering our manuscript for publication in your esteemed journal. 

We would also like to thank the reviewers for their valuable and constructive comments.

Kindly find below our point-by-point response to the academic editor and reviewers’ comments:

Academic editor

We reviewed PLOS ONE’s style requirements and made changes accordingly. 

2. PLOS requires an ORCID iD for the corresponding author in Editorial Manager on papers submitted after December 6th, 2016. 

The ORCID iD for the corresponding author was linked to the submission

3. Thank you for including your ethics statement: "The Institutional Review Board reviewed the study protocol and ethical approval for the conduction of this study was granted (IRB No. RR19393). The study was conducted under a waiver of informed consent. Data were analyzed anonymously". 

Thank you for your comment. We added a section to the manuscript titled “Ethics Statement” (Line 130-132) and included the full name of the ethics committee that approved our study. We also amended this statement in the “Ethics Statement” field of the submission system

4) We noted in your submission details that a portion of your manuscript may have been presented or published elsewhere. [poster presented at the ESCMID Conference]. Please clarify whether this [conference proceeding or publication] was peer-reviewed and formally published. If this work was previously peer-reviewed and published, in the cover letter please provide the reason that this work does not constitute dual publication and should be included in the current manuscript.

Thank you for your comment. A portion of the results discussing the trend in Ct value over time was presented as a poster presentation at the virtual ECCVID, 2020 meeting. The results have never been published elsewhere, either as part of a conference proceeding or as a peer-reviewed publication. We attached the poster presentation in the submission system for your kind reference.

Reviewers' comments:

Reviewer # 1: 

1. The present study was enrolled 461 SARS-CoV-2 infected patients who were admitted in hospital between April 4- June 5, 2020 to analyze the effect of initial viral load on the intubation and mortality of the patients and drew the conclusion that low initial viral load was related to less chances of intubation and mortality and coinciding with the descending trend of the pandemic. The data was analyzed with different but appropriate statistic ways in accord with their types, while the difference among the groups were given strong supports to the conclusion of the study. The article was written in a formal way but easy for understanding. All the tables and graphs are neat and well explained, only the image resolution is needed to get improved, especially Fig 3. Moreover, there is a missing part of sign “A” in Fig 3.

Thank you for your comments and for your observation. We split figure 3A into figure 3 and figure 4 respectively. Figure resolutions were much better maintained when graphs were included separately.

Reviewer #2: 

1. I am not familiar with the RT-PCR kit used in this study, but most of the kits that I have evaluated use internal control (IC) genes too. The Ct value of IC is a good indicator of the quality of the biological samples (with respect to collecting enough biological material as well as lacking PCR inhibitors). It is more reasonable to consider the difference between the Ct of IC and SARS-CoV-2 gene than considering the latter as an indicator of viral load. 

2. Even if the authors prefer to use absolute viral load as their main variable, comparing the IC Cts may still be helpful to exclude any variation or change in sample collection technique during this study, which may be another reason why the VLs declined over time.

Dear reviewer, 

Thank you for your feedback. We hope that the following response provides an answer to both comments #1 and #2.

The Cepheid assay used in this study does not have a control to check for specimen quality by looking for human genomic material. However, it contains a Sample Processing Control (SPC) in the cartridge utilized by the GeneXpert Xpress System instrument. The SPC is present to control for adequate processing of the sample and to monitor for the presence of potential inhibitor(s) in the RT-PCR reaction. The SPC also ensures that the RT-PCR reaction conditions (temperature and time) are appropriate for the amplification reaction and that the RT-PCR reagents are functional.

The SPC should be positive in a negative sample and can be negative or positive in a positive sample. Unfortunately, comparison between the target Ct value and specimen processing control Ct value is not practical or meaningful, as the Ct value of the SPC is impacted by the quantity of SARS CoV-2 in the patient sample and can be even negative in a positive SARS-CoV-2 test. Kindly see table from package insert ( visible in the word document uploaded with this submission)

3. Using Ct as a correlate to the viral load needs more validation. Basically, as it is well mentioned by the authors, these kits have been developed as qualitative tests. Using just 2 standard points ( in this case 10000 and 100000 VPs) is not enough to use the results as an even semi-quantitative test. I would suggest using at least 4 standard concentrations of the virus to see if the test is linear and efficient in a broader range. It is noteworthy that the observed difference between the 2 standard concentration is not 3.3 (as it is written in the manuscript) but it is 3.6 which might have larger impacts in lower or higher viral loads.

Thank you for your comment. 

We used a serial dilution of commercially available, quantified SARS-CoV-2 to better understand the performance of our Cepheid SARS-CoV-2 PCR. Our data showed a linear relationship between the viral concentration and the Ct values over four “1/10” dilutions of standard with an r2 value of 0.9.

In the manuscript, we intended to explain that theoretically, a 3.3 difference in Ct value between two concentrations occurs when the r2 value is 1.0. For r2 values less than 1.0 (eg;0.9 such as in our study), the 3.3 difference between the Ct value of two concentrations that are 10 fold different may be little higher or lower.

We clarified this point in the methods section, lines 102-106 and rearranged the format of our sentence to avoid any confusions. We also added a reference at the end of the sentence.

4. It is not clear to me how the standard virus sample has been quantified. Does it come from a virus culture or a clinical sample? Is it quantified based on genetic material copy number or based on infectious particle counting by one of virus titration methods?

Thank you for your question. We used quantified SARS-CoV-2 material for better understanding of the relationship between Ct values and the virus concentration in the patient sample. The quantified material was obtained from a commercial source. We changed the wording in the manuscript to read as: Using a commercial quantified standard, we performed studies to determine the Ct value at different target concentrations (Line 104)

5. Following the previous comments on reliability of the quantification method, in my opinion choosing the cutoffs of 26 and 36 are not well justified. We have found several COVID cases with Ct values below 20. On the other hand the intercept of the standard curve of the majority of QPCR kits is slightly higher than 35 (usually below 40), so the readouts above 35 are not usually reliable, or could be due to non-infectious viral debris.

We agree with your comment as we also have seen many positive SARS-CoV-2 cases with Ct values below 20. As a CLIA certified, CAP approved laboratory, we must follow the guidelines for reporting test result set by the manufacturer and approved by FDA for the SARSCoV-2 assay that has received FDA emergency use authorization. While there are reports that specimens positive for SARS-CoV-2 with high Ct value may be non-infectious, we still report the test as positive regardless of the Ct value. The lab leaves the interpretation of the test results with high Ct to the clinicians and medical teams. Moreover, we excluded asymptomatic patients who were admitted to the hospital for reasons other than COVID-19 infection and who were found to be COVID-19 positive on screening (for example pre-operatively) . (Fig1).

6. Based on the previous comments and uncertainties regarding the QPCR results, I would recommend statistical re-analyzing the data with parametric test as well. In those analyses the absolute value of Ct ( or delta Ct, in case the Ct of the internal controls are available) could be used. Then the author may come to similar conclusion, or find out choosing different cutoffs for stratification is more suitable, or just report the results of the parametric analysis.

Thank you for your constructive comment. We agree that re-analyzing the data by using the Ct value as a continuous variable is extremely important. We added a paragraph to the methods (lines 123-127 and results (lines 205-209) sections describing our statistical methods and results (as detailed below). We also added an extra table (Table 3) to the main manuscript as we believe that these findings strengthen our results. 

We performed a Shapiro-Wilk test of normality on the continuous variables age and Ct values. Both did not assume a normal distribution. We then proceeded to approximate their correlation independently with mortality using a Mann-Whitney U test. A multiple linear regression model was then performed. 

Age and Ct value were independently associated with in-hospital mortality (p<0.001). On multiple linear regression, both age (ß= 0.182, p<0.001) and Ct value (ß= -0.251, p<0.001) were significantly associated with mortality, such that Death = 0.431 + 0.006*Age - 0.016*Ct value. The negative association between Ct and mortality reflects the fact that lower Ct values indicates higher viral load. 

7. Based on results presented in table 1, it seems to me that Age and Viral load are rather highly correlated. In table 2 both have very close ORs on Mortality rate as well ( 5.50 vs 5.42). I wonder what is the correlation of these two variable with each other, and whether they are not covariates. Have the author tested other variables in different age groups ( or again, using age as a parametric variable against other variables)? In the Statistical Analysis section there is a reference to table S1, which I could not find it in the manuscript or its supplementary materials.

Thank you for your comment. There was an inverse association (p<0.001) between age and Ct value such that Ct value = 39.3 – 0.088*Age. These two variables do seem to be related, however, after controlling for age in our model, a higher viral load was still associated with increased mortality. Given that the main goal of our study was to analyze the association between viral load and outcomes (mortality and intubation) and explore the potential use of Ct value in predicting the course of the pandemic, we elected not to explore the relationship between Ct and age or age and other variables.

The supplementary table was not uploaded separately as part of the supplementary material (by mistake); It was instead included at the bottom of the manuscript below the references section. We re-uploaded the table separately as part of the supplementary material with this revised submission.

8. In table 1, the average level of Troponin is 40, while the average of each VL group is below 29. I wonder if it is correct.

Thank you for your observation and thorough review of the manuscript. The value was indeed incorrect. The corrected troponin value is 28 (11-80) ; this was adjusted in Table 1.

Reviewer #3: 

The manuscript describes a cohort study on relation among initial viral load in nasopharyngeal covid-19 samples and severity of the disease as well as status of pandemic. The subject is well designed and performed although some points need authors’ attention:

1. An important factor influencing the outcome of the disease and the result of this study is the type of treatment and should be considered in all interpretations. Type of antiviral or anti-inflammatory drugs, interval time between PCR and starting the treatment should be considered in analysis.

Dear reviewer,

Thank you for your constructive comment. We added a paragraph in the results section to address your comment (Lines 212-221). We did collect data on 3 medications: steroids, tocilizumab and remdesivir. At the time of the study, little evidence was available to support the use of these medications in patients with COVID-19 infection. A minority of patients received tocilizumab or remdesivir. Corticosteroids were prescribed to 52.3% (n=102) of patients with low viral load, 73.2% (n=123) of patients with intermediate viral load and 78.6% (n=77) of those with high viral load. Patients who received steroids had higher mortality rates compared to patients who did not (35.8% vs 15.1%, p < 0.011). Since corticosteroids were generally reserved for more severe cases, the increased prevalence of their use in higher viral load categories reflects the described relationship between viral load and mortality highlighted our study. No conclusions regarding the effect of corticosteroids on mortality could be drawn.

On a related note, given that we only included Ct values that were obtained on initial admission to the hospital and prior to any therapeutic interventions, the Ct values should not be affected by any type of therapy that patients received during their hospitalization.

2. An explanation of the division of weeks according to pandemic conditions should be given at the first appearance in the text.

Thank you for your comment. We added a brief sentence describing the pandemic condition in Michigan as it relates to the weeks of our study. (lines 90-94)

3- In comparing with the report of Magleby et al. in addition to the reasons given for the discrepancy with the results of the present study, the authors should also examine and discuss the pandemic conditions in two different studied time periods. Similarly, due to the fact that pandemic peaks are not the same in different countries, this point should be considered in comparison with the data of other countries.

Thank you for your comment. We added a sentence mentioning that pandemic conditions between our study and Magleby et al. were comparable as both studies spanned a period when Michigan and New York city witnessed their first peak in newly diagnosed COVID-19 cases coinciding on April 7 and April 12 respectively (Lines 271-274)

Regarding the paragraph discussing the trend in Ct values in Italy, the authors of the study did confirm that Ct values were collected during the month of April, which represented the peak of the pandemic in that region of Italy and compared these values to those obtained, in May, a time when the pandemic in Italy became less severe. This is also comparable to our study as we followed Ct values between April 4 (peak of the pandemic) until June 5, a time when the pandemic severity had been weaning down gradually.

4 -The sentence in line 97 and 98 needs reference.

Thank you for your comment; We added 2 references (now line 101)

Sincerely,

Pranatharthi Chandrasekar, MD

Division of Infectious Diseases

Wayne State University School of Medicine, Detroit, Michigan 

Phone: 313-745-7105 / Fax: 313-993-0302

---

## [Decision Letter · Decision Letter 1]

16 Jun 2021

PONE-D-21-08808R1

SARS-CoV-2 infection: Initial viral load (iVL) predicts severity of illness/outcome, and declining trend of iVL in hospitalized patients corresponds with slowing of the pandemic

PLOS ONE

Dear Dr. Chandrasekar,

Thank you for submitting your manuscript to PLOS ONE. After careful consideration, we feel that it has merit but does not fully meet PLOS ONE’s publication criteria as it currently stands. Therefore, we invite you to submit a revised version of the manuscript that addresses the points raised during the review process.

We look forward to receiving your revised manuscript.

Kind regards,

Tai-Heng Chen, M.D.

Academic Editor

PLOS ONE

Journal Requirements:

Reviewers' comments:

Reviewer's Responses to Questions

**Comments to the Author**

1. If the authors have adequately addressed your comments raised in a previous round of review and you feel that this manuscript is now acceptable for publication, you may indicate that here to bypass the “Comments to the Author” section, enter your conflict of interest statement in the “Confidential to Editor” section, and submit your "Accept" recommendation.

Reviewer #2: All comments have been addressed

Reviewer #3: All comments have been addressed

2. Is the manuscript technically sound, and do the data support the conclusions?

Reviewer #2: Partly

Reviewer #3: Yes

3. Has the statistical analysis been performed appropriately and rigorously? 

Reviewer #2: I Don't Know

Reviewer #3: I Don't Know

4. Have the authors made all data underlying the findings in their manuscript fully available?

Reviewer #2: No

Reviewer #3: Yes

5. Is the manuscript presented in an intelligible fashion and written in standard English?

Reviewer #2: Yes

Reviewer #3: Yes

6. Review Comments to the Author

Reviewer #2: Thank you for addressing all comments. However, I still can not see the evidence of any standard curve to partially validate the quantification of VL by the QPCR. IT is mentioned that 1 to 10 dilution was used to draw such a standard curve and R^2 was 0.9, but no data has been provided. The source or specifications of the commercial virus sample is not clear as well.

I am not sure, but I think the added statistical analysis is not accurate. I appreciate testing for normality of data and comparing age and Ct in outcome groups by Mann-Withney U test, but I do not understand how and on what variables the linear regression model was fitted. the equation for table 3, Death = 0.431 + 0.006*Age - 0.016*Ct value, looks like a logistic regression equation to me, whereas by "Death" the authors mean the probability of death, but it is not mentioned in the method section. Beside, in the same table, while age and Ct value are parametric variables, CKD must be nominal and it is not clear to me by which method this p value has been calculated.

Other answers are satisfactory

Reviewer #3: My comments have been addressed by the authors, but regarding the reviewer #2 comments 1 &2 it is suggested that brief explanations similar to the response to the reviewer about the Cepheid GenExpert instrument system be given in the text.

7. PLOS authors have the option to publish the peer review history of their article (what does this mean?). If published, this will include your full peer review and any attached files.

Reviewer #2: No

Reviewer #3: No

---

## [Author Response · Author response to Decision Letter 1]

17 Jul 2021

Dear editor

Tai-Heng Chen, M.D.

PLOS One 

Thank you for considering our manuscript for publication.

We would also like to thank the reviewers for their valuable and constructive comments.

Kindly find below in bold our point-by-point response to the academic editor and reviewers’ comments:

Reviewer #2: 

1. Thank you for addressing all comments. However, I still can not see the evidence of any standard curve to partially validate the quantification of VL by the QPCR. IT is mentioned that 1 to 10 dilution was used to draw such a standard curve and R^2 was 0.9, but no data has been provided. The source or specifications of the commercial virus sample is not clear as well.

Thank you for your comment. Please find below our answer to comments #1 and #2 respectively: 

Source and specifications of the commercial virus sample:

In order to determine the performance of Cepheid Genexpert SARS-CoV-2 assay we used quantified standard materials that were obtained from Exact Diagnostics (www.exactdiagnostics.com). The product from Exact diagnostics is EDX SARS-CoV-2 Standard that is manufactured with synthetic RNA transcripts containing five gene targets (E, N, ORF1ab, RdRP and S Genes of SARS-CoV-2). The product allows laboratories to validate/verify the entire process of the assay including extraction, amplification, and detection. The EDX SARS-CoV-2 Standard contains E, N, ORF1ab, RdRP and S genes that are each quantitated at 200,000 cp/mL using Bio-Rad Digital Droplet PCR (ddPCR).

The other commercially available source that we used was Zeptometrix NATtrol™ (SARS-CoV-2) External positive Control. It was used as well for the validation/verification studies and in determination of the lower limit of detection of our SARS-CoV-2 assay.

This control is a quantified source for SARS-CoV-2 and has a concentration of 50,000 copies/ml.

We also received SARS-CoV-2 virus from BEI resources repository (www.niaid.nih.gov/research/bei-resources-repository after signing a material transfer agreement. Being able to obtain the SARS-CoV-2 was a big factor in quick validation/verification of our SARS-CoV-2 assay at the very early stage of COVID-19 pandemic which allowed our laboratory to start testing patient samples for COVID-19 in March of 2020.

Standard curve:

We are attaching a picture of the standard curve in the submission system as a supplementary file and the figure can be also seen in the response to reviewers word document attached with this revision.

After repeating the experiment, we have made few changes pertaining to the correlation between the viral loads and corresponding Ct values. The changes made are highlighted below, and these do not affect our results or conclusions. (Lines 110-114)

At 2x102, 2x103 and 2x104 viral particles, the Ct values were 43, 39.4 and 36.7 respectively. We designated high, intermediate, and low VL samples to have a Ct value of ≤ 25, 26-36, and ≥ 37 consecutively, therefore, assuming a linear relationship between the Ct value and target concentration, samples with a Ct value of 26 should have a VL of 2x107 while samples with a Ct value of 36 should have a VL of 2x 104 approximately.

The viral load categories (low, intermediate and high) were chosen based on the Ct values of <25, 26-36 and > 37 respectively. Other comparable categories of viral load are used in the literature. (Rabaan AA, et al. Diagnostics (Basel). 2021;11(6):1091 and Rao SN, et al. Infect Dis Ther 2020;9(3):573-586. doi:10.1007/s40121-020-00324-3.

Correlating a Ct value to a viral load in our manuscript is used only to provide perspective to the readers. Moreover, the association between Ct value and mortality was also demonstrated on multivariate logistic regression when Ct values were analyzed as continuous variables.

2. I am not sure, but I think the added statistical analysis is not accurate. I appreciate testing for normality of data and comparing age and Ct in outcome groups by Mann-Withney U test, but I do not understand how and on what variables the linear regression model was fitted. the equation for table 3, Death = 0.431 + 0.006*Age - 0.016*Ct value, looks like a logistic regression equation to me, whereas by "Death" the authors mean the probability of death, but it is not mentioned in the method section. Beside, in the same table, while age and Ct value are parametric variables, CKD must be nominal and it is not clear to me by which method this p value has been calculated.

Based on your recommendations, we performed a multivariate logistic regression model using age and Ct value as continuous variables and added a paragraph to the manuscript. (Lines 234-240).

Using multiple logistic regression, both age and Ct value were found to be significantly associated with mortality, such as every 1-year increase in age results in 3.5% increase in odds of death and every 1 unit increase in Ct value decreases the odds of death by 8%. The negative association between Ct and mortality reflects the fact that lower Ct values indicate higher viral load (Table 3 - can be seen in the main body of the manuscript and word document of the response to reviewers attached with this submission)

Reviewer #3:

My comments have been addressed by the authors, but regarding the reviewer #2 comments 1 &2 it is suggested that brief explanations similar to the response to the reviewer about the Cepheid GenExpert instrument system be given in the text.

Thank you for your feedback. We added a paragraph to the methods section briefly explaining how our assay does not contain an internal control but instead, the SPC utilized by the GeneXpert Xpress System instrument serves to confirm adequate processing of the sample and to ensure that the RT-PCR reaction conditions are appropriate (Lines 102- 105).

Sincerely,

Pranatharthi Chandrasekar, MD

Division of Infectious Diseases

Wayne State University School of Medicine, Detroit, Michigan 

Phone: 313-745-7105 / Fax: 313-993-0302

---

## [Decision Letter · Decision Letter 2]

28 Jul 2021

SARS-CoV-2 infection: Initial viral load (iVL) predicts severity of illness/outcome, and declining trend of iVL in hospitalized patients corresponds with slowing of the pandemic

PONE-D-21-08808R2

Dear Dr. Chandrasekar,

We’re pleased to inform you that your manuscript has been judged scientifically suitable for publication and will be formally accepted for publication once it meets all outstanding technical requirements.

Kind regards,

Tai-Heng Chen, M.D.

Academic Editor

PLOS ONE

Reviewers' comments:

Reviewer's Responses to Questions

**Comments to the Author**

1. If the authors have adequately addressed your comments raised in a previous round of review and you feel that this manuscript is now acceptable for publication, you may indicate that here to bypass the “Comments to the Author” section, enter your conflict of interest statement in the “Confidential to Editor” section, and submit your "Accept" recommendation.

Reviewer #2: All comments have been addressed

Reviewer #3: All comments have been addressed

2. Is the manuscript technically sound, and do the data support the conclusions?

Reviewer #2: Yes

Reviewer #3: Yes

3. Has the statistical analysis been performed appropriately and rigorously? 

Reviewer #2: Yes

Reviewer #3: Yes

4. Have the authors made all data underlying the findings in their manuscript fully available?

Reviewer #2: Yes

Reviewer #3: Yes

5. Is the manuscript presented in an intelligible fashion and written in standard English?

Reviewer #2: Yes

Reviewer #3: Yes

6. Review Comments to the Author

Reviewer #2: Thank you for revising the statistical analysis. I have just one further recommendation, in real time PCR analyses, the standard curve graph (here presented in the supplement) is usually drawn as Log of concentration in the X axis and Ct values in the Y axis. Though the statistical meaning is the same, this is more common because in the resulting regression equation, the slope could be easily converted to the efficiency of the PCR.

Reviewer #3: (No Response)

7. PLOS authors have the option to publish the peer review history of their article (what does this mean?). If published, this will include your full peer review and any attached files.

Reviewer #2: **Yes: **Kayhan Azadmanesh MD. Ph.D.

Reviewer #3: **Yes: **Dr T Bamdad

---

## [Editor Report · Acceptance letter]

8 Sep 2021

PONE-D-21-08808R2 

SARS-CoV-2 infection: Initial viral load (iVL) predicts severity of illness/outcome, and declining trend of iVL in hospitalized patients corresponds with slowing of the pandemic 

Dear Dr. Chandrasekar:

I'm pleased to inform you that your manuscript has been deemed suitable for publication in PLOS ONE. Congratulations! Your manuscript is now with our production department. 

Kind regards, 

on behalf of

Dr. Tai-Heng Chen 

Academic Editor

PLOS ONE